

# Multiple relaxation times in perturbed XXZ chain

**Marcin Mierzejewski[1], Jakub Pawłowski[1], Peter Prelovšek[2,3] and Jacek Herbrych[1⋆]**

**1** Department of Theoretical Physics, Faculty of Fundamental Problems of Technology,
Wrocław University of Science and Technology, 50-370 Wrocław, Poland
**2** Jožef Stefan Institute, SI-1000 Ljubljana, Slovenia
**3** Faculty of Mathematics and Physics, University of Ljubljana, SI-1000 Ljubljana, Slovenia

⋆ jacek.herbrych@pwr.edu.pl

## Abstract

We numerically study the relaxation of correlation functions in weakly perturbed integrable XXZ chain. The decay of the spin-current and the energy-current correlations at zero magnetization are well described by single, but quite distinct, relaxation rates governed by the square of the perturbation strength $g$. However, at finite magnetization a single correlation function reveals multiple relaxation rates. The result can be understood in terms of multi-scale relaxation scenario, where various relaxation times are linked with various quantities which are conserved in the reference integrable system. On the other hand, the correlations of non-commuting quantities, being conserved at particular anisotropies $\Delta$, decay non-exponentially with characteristic time scale linear in $g$.



# 1 Introduction

Integrable quantum many-body systems attract a lot of attention due to their unique properties, as well as due to development of new analytical and numerical tools to deal with them (for a recent review see Ref. [1]). The crucial role in the behavior of such systems is played by the presence of extensive number of local and/or quasilocal conserved quantities (CQ), which have important consequences for the (lack of) relaxation of observables and for the transport properties. The latter consequences are formally expressed via the Mazur bounds which relate the long-time correlations (stiffness) of observables with their projections on the local CQ (charges) [2,3]. However, the microscopic models are integrable only for fine-tuned sets of parameters, while more realistic systems might be described in terms of nearly integrable (NI) models which contain small but non-vanishing perturbation that breaks the integrability [4,5]. An important question concerns the details of the integrability breaking, in particular whether the asymptotic dynamics becomes consistent with the generic dissipative diffusive-type transport [6–8]. So far, the properties of NI models are better understood at intermediate time-scales, when the dynamics resembles that of integrable models, the phenomenon known as prethermalization [9–11].

The problem of asymptotic dynamics of NI models at long time scales [12,13] appears to be more complex. It has been argued that the time evolution can be accurately described as the generalized Gibbs ensemble [14,15] with the time-dependent Lagrange parameters [16]. Several analytical and numerical studies have demonstrated that breaking of integrability leads to exponential relaxation of typical observables and that the corresponding relaxation rates scale quadratically with the strength of the perturbation [12,17–20]. Description of NI models within the framework of generalized hydrodynamics (GHD) [21–25] seems also demanding, since the generic integrability-breaking processes involve large momenta transfers [26,27]. Nevertheless, recent results suggest that arbitrarily weak perturbation applied to a macroscopic integrable system restores the generic chaotic (dissipative) dynamics [28]. Still, more detailed understanding or even theoretical analysis of the relaxation of different quantities in NI systems is mostly lacking so far.

In this work we numerically study relaxation of several operators in a NI XXZ model, employing the microcanonical Lanczos method (MCLM) [29–31] which allows to reach long-enough times even for NI model. In particular, we confirm that both spin-current and energy-current decay exponentially in time with relaxation rates that scale quadratically with the strength of perturbation. However, the energy-current relaxation rate turns out to be much smaller than the relaxation rate for the spin current. The presence of distinct relaxation times is consistent with the predictions of GHD [26,27]. This result can be simply explained by single (but different for both quantities) local/quasilocal CQ (charges) involved in the relaxation process [19]. Still even in this case, a more detailed analysis using the memory-function indicates a weak contribution of other CQ from the same symmetry sector. Moreover, we can construct a single observable that clearly reveals multiple relaxation rates or, in other words, that its relaxation cannot be described by a single exponential function. We confirm the latter possibility studying relaxation of current correlations in sectors with small magnetization, i.e., with non-zero total spin $S_{\text{tot}}^z \neq 0$. Results in this case can be explained with a projection on the decay of several CQ with different relaxation times. Furthermore, we find that the relaxation explicitly depends on the form and the symmetry of the perturbation.

Finally, we study the weakly perturbed XXZ model for the specific values anisotropy parameters, $\Delta$, when the many-body spectra are macroscopically degenerate. Such degeneracy in the integrable model allows for additional local CQ which do not commute with $S_{\text{tot}}^z$ or with other local CQ [32]. It is the case, e.g., for the anisotropy parameter $\Delta = 0.5$ which we use in the present work. However, the simplest example can be studied for $\Delta = 1$, where $S_{\text{tot}}^x$ is con-

served and but does not commute with $S^z_{\text{tot}}$. Upon introducing a perturbation, the correlation functions of quantities which do not commute with $S^z_{\text{tot}}$ show - instead of exponential decay - an approximately Gaussian decay with the characteristic relaxation time that scales linearly with the perturbation strength.

## 2 Relaxation of spin and energy currents

We consider the one-dimensional XXZ chain with $L$ sites assuming periodic boundary conditions, with a specific perturbation involving the next-nearest-neighbor interaction

$$ H = H_0 + gH', \quad H_0 = \sum_i h_i, \quad h_i = \frac{J}{2}\left(S^+_i S^-_{i+1} + S^-_i S^+_{i+1}\right) + J\Delta S^z_i S^z_{i+1}, \quad H' = J\sum_i S^z_i S^z_{i+2}, \quad (1) $$

where $S^{\pm,z}$ are spin-$\frac{1}{2}$ operators. The model is integrable for $g = 0$ and the integrability is broken for $g \neq 0$. In this section we study the relaxation of the spin current, $j_\sigma$, as well as of the energy current, $j_\kappa$, obtained for the unperturbed system

$$ j_\sigma = i\sum_{l,l'} l[h_{l'}, S^z_l], \qquad j_\kappa = i\sum_{l,l'} l[h_{l'}, h_l], \qquad (2) $$

for which we calculate the corresponding normalized current-current correlation functions

$$ C_\sigma(t) = \frac{\langle j_\sigma(t)j_\sigma\rangle}{\langle j_\sigma j_\sigma\rangle}, \qquad C_\kappa(t) = \frac{\langle j_\kappa(t)j_\kappa\rangle}{\langle j_\kappa j_\kappa\rangle}. \qquad (3) $$

We focus on the result for large (infinite) temperature $T = 1/\beta \gg J$. Here, $\langle\dots\rangle = \text{Tr}(\dots)$ denotes averaging either over the canonical ensemble with fixed $S^z_{\text{tot}}$ (in Secs. 2-4) or grand-canonical ensemble (in Sec. 5), and $j_\alpha(t) = e^{iHt}j_\alpha e^{-iHt}$. Note that $C_\sigma(t)$ and $C_\kappa(t)$ are related via the Fourier transform to the dynamical spin conductivity, $\sigma(\omega) = \beta\langle j_\sigma j_\sigma\rangle\tilde{C}_\sigma(\omega)$, and the thermal conductivity, $\kappa(\omega) = \beta^2\langle j_\kappa j_\kappa\rangle\tilde{C}_\kappa(\omega)$, respectively.

For the numerical evaluation of the spectral functions $\tilde{C}_\alpha(\omega)$, on systems up to $L = 28$ sites, we use the MCLM which allows for very high frequency resolution, $\delta\omega \lesssim 10^{-3}J$, and consequently enables evaluation of $C_\alpha(t)$ up to $t \lesssim t_{\max} \sim 10^3/J$. Reaching large $t_{\max}J \gg 1$ is crucial to follow the slow relaxation at weak perturbations, $g \ll 1$, and to resolve possible distinct relaxation times. This is achieved within MCLM by using large number of Lanczos steps $M_L \sim 5.10^4$. In the following, we numerically calculate canonical $C_\alpha(t)$, i.e., in sectors with fixed total spin projections $S^z_{\text{tot}}$. Large but finite $M_L$ also sets the smallest reliable value of $C_\alpha(t) \gtrsim 2.10^{-3}$.

As a first step, we establish the range of the perturbation strength, $g$, which is relevant for NI systems. On the one hand, $g$ should be small to remain a perturbation but, on the other hand, $g$ should be sufficiently large so that the effective mean-free path is smaller then the system size ($L \leq 28$). The latter requirement means that the accessible system sizes impose lower bounds on the accessible values of $g$. Since we are not able to directly evaluate the mean-free path, we use the range of parameters for which the decay of the correlation functions does not reveal any significant $L$ dependence but also exhibits a clear quadratic dependence on $g$ [12, 19, 20]. Main panels in Fig. 1(a,b) show, respectively, $C_\sigma(t)$ and $C_\kappa(t)$ within the sector $S^z_{\text{tot}} = 4$ for various $g = 0.1 - 0.4$, whereas the time in the insets is rescaled by $g^2$. For $0.15 \leq g \leq 0.3$ we observe a perfect collapse of all curves, whereas the latter collapse is worse for weaker $g$ indicating that the effective mean-free path becomes larger than the systems size. Therefore from now on, we set $g = 0.15$.

In Fig. 2(a) we directly compare the relaxation of spin and energy currents $C_\alpha(t)$ at zero-magnetization, $S^z_{\text{tot}} = 0$. Here, $H$ is invariant under the $Z_2$ spin-flip transformation, generated

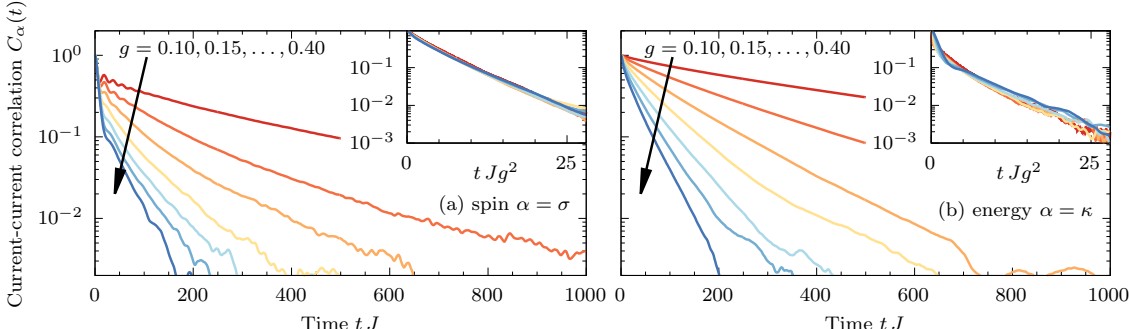

Figure 1: Normalized correlation functions $C_\alpha(t)$ for (a) spin current $j_\sigma$ and (b) energy current $j_\kappa$, respectively, in $S^z_{\text{tot}} = 4$ sector, calculated for $L = 28$ and $\Delta = 0.5$. Insets: the same as in main panels but with the time $t$ rescaled by the square of the perturbation strength $g^2$.

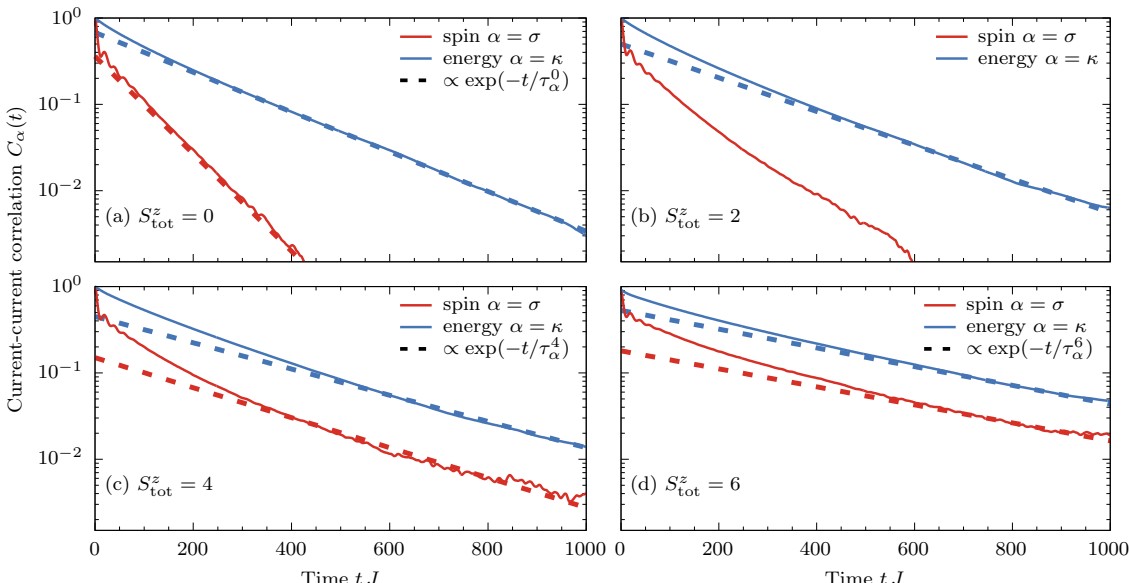

Figure 2: Correlation functions $C_\alpha(t)$ in sectors with various $S^z_{\text{tot}}$. Dashed curves in (a) depict exponential fits for $\alpha = j_\sigma, j_\kappa$ at $s = S^z_{\text{tot}} = 0$, while in (c,d) for $s \neq 0$ only long-time fits are presented. Calculated for $L = 28$, $\Delta = 0.5$, and $g = 0.15$.

by the parity operator, $P = \prod_J (S^+_j + S^-_j)$. Since the spin/energy current is odd/even under this transformation, $P j_\sigma P = -j_\sigma$ and $P j_\kappa P = j_\kappa$, in this sector both currents are mutually orthogonal, i.e., $\langle j_\sigma j_\kappa \rangle = 0$. Consequently, relaxation of both correlations are evidently different. At longer times, $tJ > 100$, the decays do not show any clear deviations from simple exponential, $C_\alpha(t) \propto \exp(-t/\tau^0_\alpha)$, $\alpha = \sigma, \kappa$, where the upper index in the relaxation times, $\tau^s_\alpha$, marks the value of $s = S^z_{\text{tot}}$. Nevertheless, it is clear that the relaxation of $j_\sigma$ is much faster than that of $j_\kappa$, $\tau^0_\sigma \simeq 3\tau^0_\kappa$, confirming that the breaking of integrability leads to multiple distinct relaxation times, here due to different symmetry sectors involved. The evident differences are also at short times $t$. Since $j_\sigma$ is not CQ in the reference $H_0$, there is an incoherent drop to $C^0_\sigma \sim 0.5$ at $tJ \sim O(1)$, reflecting the nontrivial spin stiffness [3] and finite overlap with quasilocal CQ [33, 34]. On the other hand, $j_\kappa$ is CQ at $g = 0$, so one might expect a single exponential decay in the whole range of $t$. To good approximation this is indeed the case, but

there is still some visible deviation at $tJ < 100$ about which we comment in more detail in Sec. 3.

More challenging question is whether distinct relaxation rates can be observed in the dynamics of a single observable. This might happen when we consider $S_{\text{tot}}^z \neq 0$ sectors where the above symmetry arguments do not apply. In Figs. 2(b-d) we show that a departure from a simple exponential relaxation becomes increasingly visible for $S_{\text{tot}}^z \neq 0$, when also $\langle j_\sigma j_\kappa \rangle \neq 0$. Figs. 2(c,d) for $S_{\text{tot}}^z = 4, 6$ confirm that for longest $tJ > 500$ the relaxation is asymptotically determined by the same $\tau_\kappa^s$ for both currents, while $S_{\text{tot}}^z = 2$ case on Fig. 2(b) is marginal due to fast decay of $C_\sigma(t)$ (and the limitation $C_\sigma(t) > 2.10^{-3}$). Nevertheless, both correlation functions, and in particular $C_\sigma(t)$, reveal a clear deviation from a single-exponential decay.

The modest dependence of $\tau_\kappa^s$ on $s$, as extracted from Fig. 2(a-d), is shown in Fig. 3(a). More importantly, our analysis indicate that one can fit the decay of correlations $C_\alpha(t)$ for $S_{\text{tot}}^z > 0$ by a sum of two exponential functions with distinct relaxation rates $\tau_\kappa^s, \tau_\sigma^s$. This is shown in Fig. 3(b,c) for $C_\sigma(t)$, but also for $C_\kappa(t)$ in Fig. 3(d). It is indicative that the fit is consistent with only two relaxation times for both correlations, i.e., with longer relaxation rate $\tau_\kappa^s$ still weakly dependent on $s$ as given in Fig. 3(a), and much faster $\tau_\sigma^s \sim \tau_\sigma^0$ which we can approximate just with the $s = 0$ result for $C_\sigma(t)$ in Fig. 2(a).

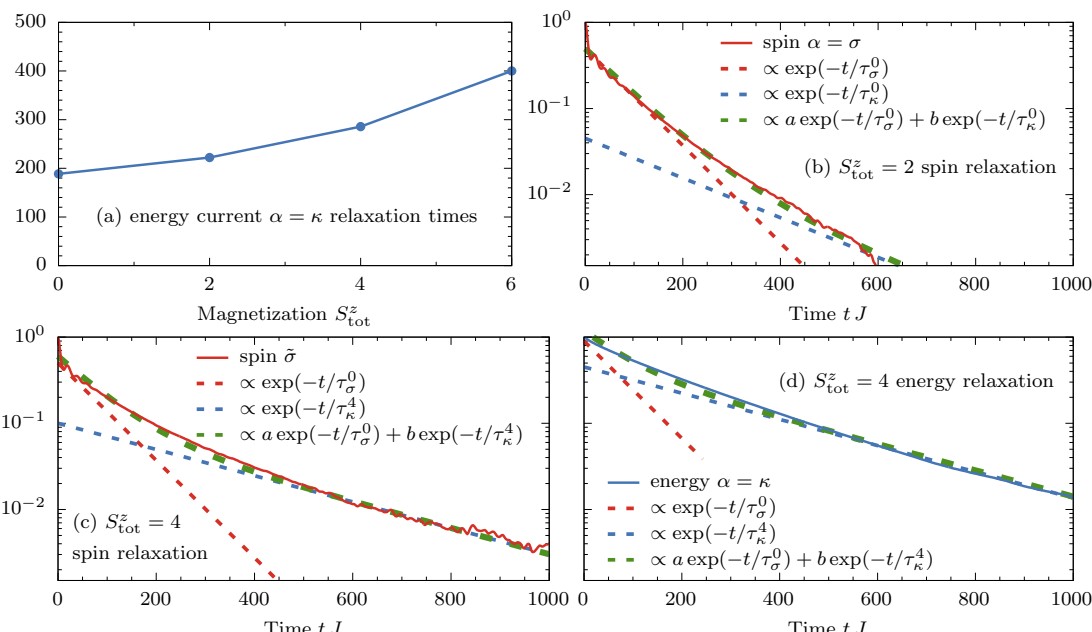

Figure 3: (a): Relaxation time for the energy current obtained from long-time fits marked by dashed lines in Fig. 2. (b,c): Continuous line shows spin-current $C_\sigma(t)$ for $S_{\text{tot}}^z = 2, 4$, respectively, while (d) energy-current $C_\kappa(t)$ for $S_{\text{tot}}^z = 4$. The results in (b,d) are fitted by a sum of two exponential functions (dashed, green curves) with the relaxation times $\tau_\kappa^s$ as in panel (a), and $\tau_\sigma^s = \tau_\sigma^0$. Calculated for $L = 28$, $\Delta = 0.5$, and $g = 0.15$.

The above results may serve as a motivation for a simple phenomenological description. The value of the correlation functions for $t \to \infty$ in the integrable model, $g = 0$, is determined via the Mazur bound [2, 3] by the projections of the studied operators on local/quasilocal CQ [33–39] $Q_n$, which should be chosen as mutually orthogonal $\langle Q_n Q_{n'} \rangle \propto \delta_{n,n'}$ in the canonical ensemble with fixed $S_{\text{tot}}^z$. Assuming the completeness (saturation) of the bound, this means

for considered correlations,

$$C_\alpha(t \to \infty) = C_\alpha^0 = \frac{1}{\langle j_\alpha j_\alpha \rangle} \sum_n \frac{\langle j_\alpha Q_n \rangle^2}{\langle Q_n Q_n \rangle}, \tag{4}$$

where the term $\langle j_\alpha j_\alpha \rangle^{-1}$ arises from normalization in Eq. (3). Eq. (4) is invariant under the orthogonal transformation of normalized CQ

$$\frac{Q_n}{\sqrt{\langle Q_n Q_n \rangle}} = \sum_s (\hat{O})_{ns} \frac{Q_s}{\sqrt{\langle Q_s Q_s \rangle}}, \tag{5}$$

where $\hat{O}$ is arbitrary orthogonal matrix. In NI system, $Q_n$ are not any more CQ and decay with the characteristic times $\propto g^2$. Based on the results in Fig. 3(b-d) and previous numerical studies in Ref. [19], we conjecture that projections on $Q_n$ are essential also for the asymptotic dynamics in NI models,

$$C_\alpha(t \gg J^{-1}) \simeq \frac{1}{\langle j_\alpha j_\alpha \rangle} \sum_n \frac{\langle j_\alpha Q_n \rangle^2}{\langle Q_n Q_n \rangle} \exp\left(-\frac{t}{\tau_n}\right), \tag{6}$$

where $\tau_n \propto 1/g^2$. It should be, however, stressed that in contrast to Eq. (4) the appropriate set of $Q_n$ in Eq. (6) is not arbitrary, but determined by the perturbation [5,17,27]. A numerical algorithm for identifying such modes has been discussed in Ref. [19]. It amounts in finding eigenvector of the matrix $M_{AB} = \langle A(t)B \rangle$ calculated in the long-time regime, $t \gg 1$, for a set of local orthogonal operators, $A$ and $B$. Since the number of local operators supported on $M$ sites grows exponentially with $M$, this method is numerically demanding. Additionally, it requires full diagonalization of the Hamiltonian hence it is applicable only to small systems. Therefore in this work, we do not attempt to determine relevant $Q_n$ in more detail.

However, for $S_{\text{tot}}^z = 0$ one can assume that one of $Q_n$ in Eq. (6) can be well approximated by $j_\kappa$ (being one of CQ in the reference system), which explains nearly perfect exponential decay of $C_\kappa(t)$ in Fig. 2(a), even though a small correction might be needed (as tested in more detail in Sec. 3). It is well known [3] that in the integrable model ($g = 0$) the spin-current stiffness, $C_\sigma^0$, is nonzero for $S_{\text{tot}}^z = 0$ provided that $\Delta < 1$. Since $\langle j_\sigma j_\kappa \rangle = 0$ for $S_{\text{tot}}^z = 0$, $Q_n$ that are relevant for the decay time $\tau_\sigma^0$ have to be related to the quasilocal CQ [19,33,34,40].

On the other hand, for nonzero $S_{\text{tot}}^z$, we clearly need at least two distinct relaxation times $\tau_\kappa^s, \tau_\sigma^s$ to fit both decays, $C_\alpha(t)$. The necessity of multiple relaxation times is most evident for $C_\sigma(t)$ at $S_{\text{tot}}^z = 2, 4$ presented on Fig. 3(b-c). Upon changing $S_{\text{tot}}^z$, one has also to modify $\tau_\kappa^s$, as suggested by the results in Fig. 3(a), while the dependence of $\tau_\sigma^s$ on $S_{\text{tot}}^z$ is less evident, so we can approximate $\tau_\sigma^s \sim \tau_\sigma^0$. In Fig. 3(d) we show that $C_\kappa(t)$ can also be well fitted by a sum of two exponential functions, with the very same relaxation times which are used for fitting $C_\sigma(t)$ in Fig. 3(c). This indicates that the same pair of $Q_n$ is relevant for the decay of the spin and the energy currents for $S_{\text{tot}}^z \neq 0$.

# 3 Memory-function analysis of the energy-current correlations

It is desirable to have more explicit way to evaluate the long-time decay of correlations, $C_\alpha(t)$, for chosen perturbations $H'$, and in particular a direct expression for relevant relaxation rates $1/\tau_n$. Here, it is convenient to follow the Mori formalism [41,42], where one can generally express the current-correlation relaxation function $\phi_\alpha(\omega)$ in terms of the corresponding memory

functions (MF), $M_\alpha(\omega)$,

$$
\begin{aligned}
\phi_\alpha(\omega) &= \frac{1}{L}\left(j_\alpha\left|[\mathcal{L}-\omega]^{-1}\right|j_\alpha\right) = \frac{\chi_\alpha(\omega)-\chi_\alpha^0}{\omega} = \frac{-\chi_\alpha^0}{\omega+M_\alpha(\omega)}, \\
\chi_\alpha(\omega) &= \frac{i}{L}\int_0^\infty e^{i\omega^+ t}\langle[j_\alpha^\dagger(t),j_\alpha]\rangle \mathrm{d}t, \qquad \chi_\alpha^0 = \chi_\alpha(0)>0,
\end{aligned}
\tag{7}
$$

where $(A|B) = 1/L\int_0^\beta \mathrm{d}\tau\langle AB(i\tau)\rangle$. At high temperatures, $\beta\to 0$, $\beta\tilde{C}_\alpha(\omega)=\mathrm{Im}\,\phi_\alpha(\omega)$ and $\chi_\alpha^0 = \beta\langle j_\alpha j_\alpha\rangle$. Knowing numerical result for $\tilde{C}_\alpha(\omega)$ at finite $g$, one can evaluate (via Kramers-Kronig relation) the complex $\phi_\alpha(\omega)$, extract directly the corresponding complex MF, $M_\alpha(\omega)$, and in particular the dynamical relaxation rate $\Gamma_\alpha(\omega)=\mathrm{Im}\,M_\alpha(\omega)$.

On the other hand, for a NI system one can find an explicit expression in the case where the current is a CQ in the reference system [17,19]. This is particularly the case for the energy current, $[H_0,Q_3]=[H_0,j_\kappa]=0$. Then, within the lowest order in the perturbation $g\ll 1$ one can approximate MF as

$$
M_\kappa(\omega)=\frac{1}{\chi_\alpha^0}N_\kappa(\omega), \qquad N_\kappa(\omega)=g^2(\mathcal{F}|[\mathcal{L}-\omega]^{-1}|\mathcal{F}), \qquad \mathcal{F}=[H',j_\kappa],
\tag{8}
$$

i.e., as the correlation function of the the force $\mathcal{F}$ in *the unperturbed - integrable system*. Eq. (8) is derived under the assumption that only a single charge $Q_3=j_\kappa$ is relevant for the decay, as well as that $\mathcal{F}$ has no overlap with any of $Q_n$. This can be true for the sector with $S_\mathrm{tot}^z=0$ and can be directly tested for operator $\mathcal{F}$ for the particular $H'$ in Eq. (1),

$$
\mathcal{F}=i\sum_i\left[T_{i-1}^{i+1}S_i^z\left(S_{i+3}^z-S_{i-3}^z\right)-\Delta T_i^{i+1}\left(S_{i+2}^z+S_{i-1}^z\right)\left(S_{i+3}^z-S_{i+2}^z\right)\right],
\tag{9}
$$

with $T_i^l=(S_l^+S_i^-+\mathrm{H.c.})/2$. Due to the parity symmetry at $S_\mathrm{tot}^z=0$, the above $\mathcal{F}$ appears orthogonal to known local/quasilocal CQ. We can then evaluate numerically $M_\kappa(\omega)$ as the correlation of $\mathcal{F}$ in the reference $H_0$ system. The MCLM result for $\Gamma_\kappa(\omega)/g^2$ at $S_\mathrm{tot}^z=0$ is presented in Fig. 4, both as extracted directly via Eq. (7) from $\tilde{C}_\kappa(\omega)$ for various finite $g=0.1-0.4$, as well as by calculating the result from Eq. (8) for the force given in Eq. (9). The overall agreement of perturbation result with the numerically extracted MF $\Gamma_\kappa(\omega)/g^2$, (being essentially $g$-independent) is very satisfactory in the whole $\omega$ regime [see the inset of Fig. 4(a)]. Still, at low $\omega/J<0.4$ there appears a visible difference. Partly responsible is the (apparently) singular contribution in perturbative $\Gamma_\kappa^0=\Gamma_\kappa(\omega\to 0)$, which seems to indicate a small overlap with some CQ (possibly being a finite-size effect). Apart from that, also visible is a quantitative mismatch at $\omega\to 0$ which indicates that the relevant $Q_n$ in Eq. (6) is not just $j_\kappa$, and consequently also Eq. (8) is not a full description of relaxation. In other words, even for $S_\mathrm{tot}^z=0$ more than a single mode is needed in Eq. (6) to properly describe $C_\kappa(t)$.

In Fig. 4(b) we present $\Gamma_\kappa(\omega)/g^2$ for fixed $g=0.15$ but various $S_\mathrm{tot}^z$. It is indicative that for $\omega/J>0.15$ the MF is essentially independent of $S_\mathrm{tot}^z$. On the other hand, the decrease of $\Gamma_\kappa^0=1/\tau_\kappa^s$ with $S_\mathrm{tot}^z$ reflects the observed increase of $\tau_\kappa^s$ in Fig. 3(a). Still, it is not straightforward to capture this in a perturbative approach, Eq. (8).

## 4 Dependence on the form of perturbation

In Sec. 2 we analysed a particular form of perturbation, i.e., the next nearest-neighbor interaction, Eq. (1), which does not break the translational symmetry or the spin parity $P$. This is also essential for the phenomenological explanation, Eq. (6), in terms of decaying CQ, as well

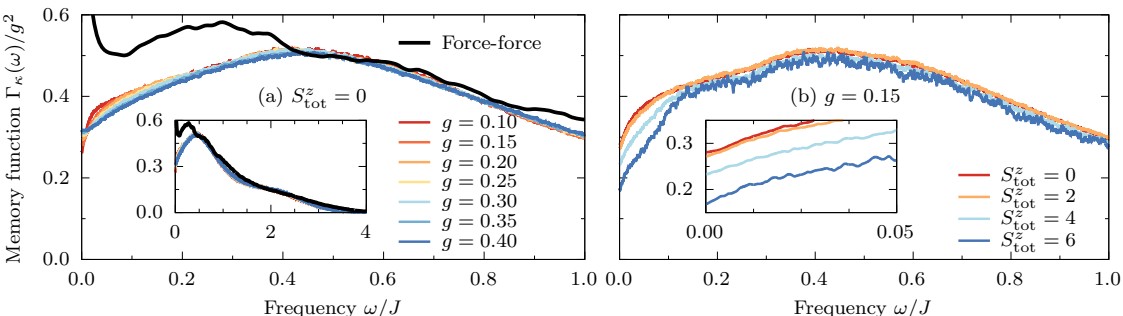

Figure 4: Energy-current memory function (relaxation-rate), $\Gamma_\kappa(\omega)$, (a) extracted directly from $\tilde{C}_\kappa(\omega)$ for $S_{\text{tot}}^z = 0$ with different perturbations $g = 0.1-0.4$, compared with the perturbation result, Eqs. (8),(9), and (b) extracted for various $S_{\text{tot}}^z$ from $\tilde{C}_\kappa(\omega)$ for fixed $g = 0.15$. Calculated for $L = 28$ and $\Delta = 0.5$.

as for the MF analysis in Sec. 3. However, using numerical MCLM we can check also other perturbations. Let us consider as a perturbation

$$H'' = J \sum_i S_i^z S_{i+1}^z S_{i+2}^z \,, \tag{10}$$

which breaks the parity symmetry, $P$, of the total Hamiltonian, $H = H_0 + gH''$. Fig. 5 shows current $\alpha = \sigma, \kappa$ correlation functions $C_\alpha(t)$ for $S_{\text{tot}}^z = 0$. We observe exponential decay with a single relaxation time $\tau_\alpha \propto 1/g^2$, which is different from two distinct relaxation times obtained for the $P$-preserving $H'$, Eq. (1), i.e., the relaxation evidently depends on the form of perturbation. One may interpret this behavior in terms of conjecture, Eq. (6). All observables which have non-vanishing projection on $Q_n$ with the longest relaxation time should asymptotically decay with the same decay rate. In order to obtain different rate, one needs to build an operator that is strictly orthogonal to the latter $Q_n$. It is highly nontrivial task, since approximate $Q_n$ in Eq. (6) can be obtained numerically only for small system. For the parity-preserving perturbations, the total Hamiltonian is even and all $Q_n$ in Eq. (6) have well defined parity being either odd or even. Then operators from one parity sectors are strictly orthogonal to $Q_n$ from the other parity sector. Due to this orthogonality we observe different relaxation times in odd and even sectors without any fine-tuning of the studied observables. However for odd perturbations, the total Hamiltonian contains even and odd terms, hence both parity sectors are mixed during the time evolution. As a consequence, $Q_n$ in Eq. (6) may not have well defined parity.

## 5 Non-commuting conserved quantities and their non-exponential relaxation

In this section, we focus on specific values of the anisotropy parameter, $\Delta$, where the many-body spectra exhibit additional massive degeneracies [31]. The latter originate from eigenstates corresponding to different $S_{\text{tot}}^z$ with equal energies. This property allows for the presence of non-commuting local/quasilocal CQ [32,43–45]. As a first example we take $\Delta = 1$ in which case $H_0$ is the SU(2)-symmetric Heisenberg chain for which one can study the total $S_{\text{tot}}^x$ spin operator

$$O_1 = \sum_j (S_j^+ + S_j^-) \,. \tag{11}$$

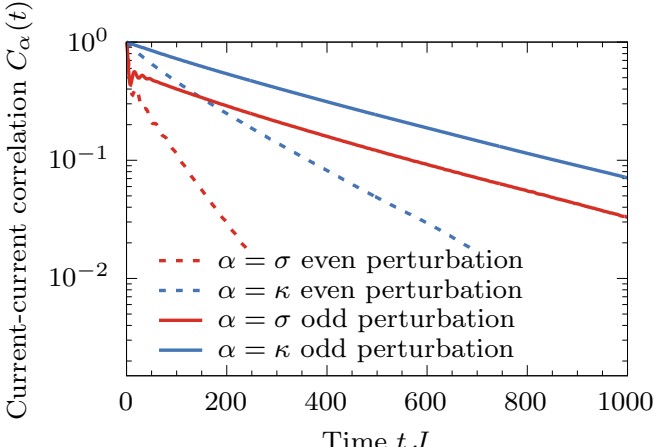

Figure 5: Current correlation function $C_\alpha(t)$ for the perturbation, Eq. (10), breaking the parity symmetry $P$ (odd perturbations), as compared with $P$-preserving perturbation Eq. (1) (even perturbations). Calculated for $L = 28$, $\Delta = 0.5$, $g = 0.15$, and $S^z_{\text{tot}} = 0$.

It is clear that $O_1$ is a local operator which commutes with $H_0$ at $\Delta = 1$, but does not commute with other local CQ, e.g., not with the $S^z_{\text{tot}}$.

The second nontrivial case is related with the commensurate $\Delta = \cos(\pi/3) = 1/2$, for which the non-commuting local CQ have been derived (for $g = 0$) in Ref. [32]. Here, we study the relaxation of

$$O_3 = \sum_j (-1)^j (S^+_{j-1} S^+_j S^+_{j+1} + \text{H.c.}), \tag{12}$$

which does not commute with $S^z_{\text{tot}}$ and is not invariant under translations by odd number of sites. It should be mentioned that analogous local operators exist for other commensurate cases, in particular for $\Delta = \cos(\pi/2) = 0$, where $O_2 = \sum_j (-1)^j (S^+_j S^+_{j+1} + \text{H.c})$ is local CQ for $H_0$ [45].

In similarity to Eq. (3), we calculate normalized correlations functions $C_1(t)$ and $C_3(t)$ for the operators $O_1$ and $O_3$, respectively. In contrast to the preceding section, now the averaging $\langle\ldots\rangle$ is carried out over grand canonical ensemble. We note that $O_3$ is similar to the previously studied spin-current $j_\sigma$ in the sense, that it does not commute with $H_0$, but has large projection on the corresponding quasilocal CQ [32]. In order to confirm this, in the inset in Fig. 6(c) we show the finite-size scaling of the relevant stiffness $\tilde{C}_3(\omega \to 0^+, g = 0)$ (with value $\simeq 0.4$ thermodynamic limit $L \to \infty$). Without imposing the translational symmetry, we studied all local non-commuting operators $\sum_i (A_i + A_i^\dagger)$, where $A_i$ are supported on up to 3 sites and do not commute with $S^z_{\text{tot}}$. Utilizing the alhorithm from Ref. [40], we have found (for $\Delta = 1/2$, $g = 0$ and $L \leq 14$) that $O_3$ has the largest stiffness out of all these operators (not shown).

Next, we focus on the asymptotic decay of $C_\alpha(t)$, $\alpha = 1, 3$, and their dependence on $g \neq 0$. To this end, we first calculate the Fourier transform, $C_\alpha(\omega)$, and then its cumulative spectral function

$$\tilde{C}_\alpha(\omega, g) = \int_{-\omega}^{\omega} d\omega'\, C_\alpha(\omega') = \frac{\sum_{mn} \theta(\omega - |E_m - E_n|)\langle m|O_\alpha|n\rangle^2}{\sum_{mn} \langle m|O_\alpha|n\rangle^2}, \tag{13}$$

expressed in terms of eigenstates, $H|n\rangle = E_n|n\rangle$, obtained using the exact diagonalization of $H$ on up to $L = 16$ sites. We note that $\tilde{C}_1(\omega \to 0^+, g = 0) = 1$ (because $O_1$ commutes with $H_0$ at $\Delta = 1$), whereas $\tilde{C}_3(\omega^+ \to 0, g = 0) \simeq 0.4$. Since we are interested in the low-$\omega$ part of

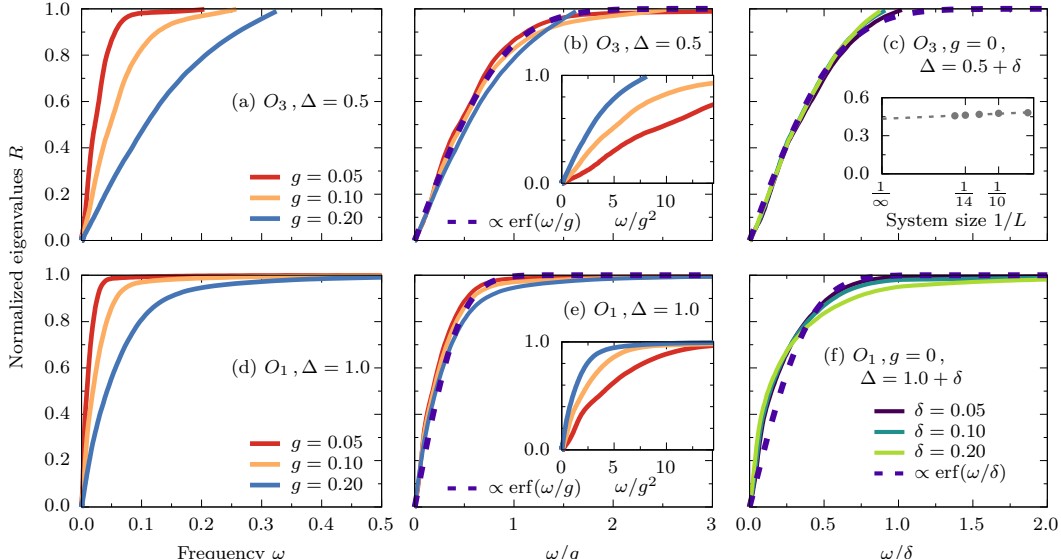

Figure 6: Cumulative spectral functions $R_\alpha(\omega)$, Eq. (14), as calculated for $L = 16$ system. We show the results for (a) $O_3$ with $\Delta = 1/2$ and (d) for $O_1$ with $\Delta = 1$. Panels (b) and (e) show respectively the same data but with rescaled frequency $\omega/g$ (main panels) and $\omega/g^2$ (insets). Panels (c) and (f) show the results for integrable system ($g = 0$) but for shifted anisotropy parameters $\Delta = 0.5 + \delta$ and $\Delta = 1 + \delta$, respectively. Dashed curves show fittings with the error function. Inset in (c) shows finite-size scaling of the stiffness $C_3(t \to \infty)$ for $\Delta = 0.5$ and $g = 0$.

$C_\alpha(\omega)$, it is convenient to normalize the spectral function,

$$R_\alpha(\omega) = \frac{\tilde{C}_\alpha(\omega, g)}{\tilde{C}_\alpha(\omega \to 0^+, g = 0)} , \qquad (14)$$

so that one may directly compare relaxations of both considered observables. The numerical results are shown in figures Figs. 6(a,b) and 6(d,e) for $R_3(\omega)$ at $\Delta = 1/2$ and $R_1(\omega)$ at $\Delta = 1$, respectively. In contrast to the spin- and energy-currents discussed in the preceding sections, curves for various $g$ do not collapse when one rescales the frequency by $g^2$, as it is shown in the insets in Fig. 6(b,e). However, a convincing collapse may be obtained for the scaling $\omega/g$. Moreover, the rescaled curves can be quite accurately fitted by the error function, shown as the dashed curves in Fig. 6, implying $\tilde{C}_\alpha(\omega) \propto \exp[-a\,(\omega/g)^2]$ where the coefficient $a$ does not depend on $g$. Consequently, the decay is not exponential but Gaussian, $C_\alpha(t) \propto \exp[-(t/\tau_\alpha)^2]$, where the characteristic relaxation rate, $1/\tau_\alpha \propto g$.

The Gaussian relaxation occurs in the vicinity of $\Delta$ characterized by additional degeneracies [31, 45], originating from that eigenstate with different $S_{\text{tot}}^z$ having the same energy. The perturbation breaks integrability but also lifts the latter degeneracy. In order to disentangle these two mechanisms, we also study the correlation functions, $\tilde{C}_\alpha(\omega, g)$ for $g = 0$ but with shifted $\Delta = 1/2 + \delta$ (for $\alpha = 3$) and $\Delta = 1 + \delta$ (for $\alpha = 1$). Then, the degeneracy is lifted without destroying the integrability. The results are shown in Fig. 6(c,f) for both $\alpha = 1, 3$. One observes the same behavior as for the NI system above, i.e., $C_\alpha(t) \propto \exp[-(t/\tau_\alpha)^2]$, with $1/\tau_\alpha \propto \delta$. This result explains additionally the origin of the anomalous scaling of characteristic $\tau_\alpha$. The nonzero stiffnesses, $\tilde{C}_\alpha(\omega \to 0^+, g = 0)$, emerges from states $|m\rangle$ and $|n\rangle$ in Eq. (13) with different $S_{\text{tot}}^z$, but with equal energies $E_m = E_n$. The latter degeneracy is lifted

either by $g \neq 0$ or $\delta \neq 0$ already in the first order perturbation theory, $|E_m = E_n| \propto g, \delta$ [45], leading to the scaling $\omega/(g, \delta)$, shown in Fig. 6.

## 6 Conclusions

We numerically analyzed the decay of normalized correlation function $C_\alpha(t)$ of different local quantities in the nearly integrable XXZ model. Correlations generally reveal a fast drop at short times $tJ \sim O(1)$, consistent with finite stiffnesses of studied quantities in the integrable model $H_0$. A weak integrability breaking $g \ll 1$ then leads to further slow - exponential-like - decay which is characterized with a single or multiple relaxation times, all scaling with the perturbation strength $\tau_n \propto 1/g^2$. The simplest cases appear to be the energy-current $j_\kappa$ and spin-current $j_\sigma$ correlations at zero magnetization $S_{\text{tot}}^z = 0$, where - due to symmetry - different CQ are involved in the relaxation of both quantities, and consequently relevant $\tau_n$ are quite distinct. Since $j_\kappa$ is by itself CQ within $H_0$, one can go a step further and give an explicit memory-function analysis and perturbative expression for the relaxation-rate spectral function $\Gamma_\kappa(\omega)$. In this case the extracted and perturbative $\Gamma_\kappa(\omega)$ match quite well in the whole $\omega$ range. Still, some deviations in $\Gamma_\kappa(\omega \sim 0) = 1/\tau_\kappa^0$ as well as in $C_\kappa(t)$ at shorter $t/g^2$ indicate on possible multiple relaxation times and more than one relevant $Q_n$ even in this case.

The existence of multiple relaxation times is becoming evident for $S_{\text{tot}}^z \neq 0$. The conjecture Eq. (6) concerning the presence of multiple relaxation times $\tau_n$, which are linked with (appropriately rotated) various CQ of the parent integrable model, is not in conflict with previous results [20] which report a simple exponential relaxation. In order to demonstrate at least two relaxation times, we have carefully selected observables so that they have a large projection on quickly decaying CQ for $S_{\text{tot}}^z \neq 0$ ($Q_n$ relevant for $j_\sigma$ at $S_{\text{tot}}^z = 0$), as well as much smaller projection on slowly decaying one ($Q_3 = j_\kappa$). Namely, we made us of the result from Ref. [3] that the projection $\langle j_\sigma j_\kappa \rangle$ is proportional to $S_{\text{tot}}^z$ thus the projection may be easily tuned via changing the magnetization sector. Typically, the opposite holds true: one studies observables which have largest projection on simplest CQ, which are supported only on the few sites and the related $\tau_n$ are the longest relaxation times [19]. As a consequences, small and fast decaying projections on more complicated CQ in Eq. (6) may not be visible in the numerical results. Such generic scenario of the multi-scale relaxation is consistent with numerical results obtained in the present work as well as in Ref. [19]. Nevertheless, it should be considered as conjecture that requires further studies and should be verified for other nearly integrable systems.

Furthermore, our analysis indicates that the form, in particular the symmetry, of the perturbation is relevant for the decay of $C_\alpha(t)$. In contrast to quantities considered above at general anisotropy $\Delta$ and their decay, the correlations $C_\alpha(t)$ of particular quantities $O_l$, being conserved by $H_0$ only at commensurate $\Delta_0 = \cos(\pi/m)$, but not commuting with $S_{\text{tot}}^z$, behave qualitatively different. Under finite perturbation, which here can be introduced either by $g \neq 0$ or by deviation $\Delta = \Delta_0 + \delta$, we observe effectively a (non-exponential) Gaussian-like decay of $C_\alpha(t)$ with different scaling of characteristic decay time, i.e. $\tau_\alpha \propto 1/g$. The origin here is the lifting of macroscopic degeneracy of many-body states at these particular $\Delta_0$.

**Funding information** J.H. acknowledges the support by the Polish National Agency of Academic Exchange (NAWA) under contract PPN/PPO/2018/1/00035. M.M. acknowledges the support by the National Science Centre, Poland via projects 2020/37/B/ST3/00020. P.P. acknowledges the support by the project N1-0088 of the Slovenian Research Agency. The numerical calculation were partly carried out at the facilities of the Wroclaw Centre for Networking and Supercomputing.

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
