# Peer review of "Multiple relaxation times in perturbed XXZ chain"

_SciPost Physics, doi:SciPost Phys. 13, 013 (2022)_

## Round 1 · Referee Report · Anonymous (Referee 1) · 2022-4-13

Strengths

-Quality numerical data and careful analysis.
-The authors address a challenging problem.

Weaknesses

-No strong conclusion based on the data.

Report

In the paper the authors address the relaxation in weakly perturbed integrable systems. Precisely, they study the XXZ spin chain with next-nearest neighbor interactions. They show that while in the zero magnetization sector the spin and current correlation functions decay exponentially with a single time scale, in other magnetization sectors they are better described by the sum of two exponentials. In the paper they provide several arguments justifying this behavior based on the structure of the conserved quantities of the XXZ chain.

Now, this paper provides a very careful analysis of quality exact diagonalization data for a very challenging problem such as that of relaxation dynamics in close to integrable systems. The results are quite valuable and can be useful to clarify the underlying scenario. Unfortunately, despite the careful analysis it is difficult to reach a conclusion on the proposed multi-scale relaxation scenario. This is the main weak point of the paper. As this is due to the intrinsic challenging nature of the problem, I am inclined to recommend the paper for publication in Scipost Physics. My main criticism about the approach is the following. The authors study numerically chains with L=25 sites up to times of order 10^3. One would expect that to address relaxation one should consider times t\lesssim L, avoiding revival effects. I would appreciate if the authors could address and clarify this point.

Requested changes

At page 2 at the beginning of the second paragraph there is a typo : ensable should be ensemble.

  • validity: high
  • significance: good
  • originality: good
  • clarity: good
  • formatting: excellent
  • grammar: excellent

Author:  Jacek Herbrych  on 2022-05-21  [id 2501]

(in reply to Report 1 on 2022-04-13)

We thank the Referee for pointing out this unclear point of our presentation. We have formulated our conjecture for the multi-scale relaxation scenario at the end of the second paragraph of conclusions that starts with “Typically, the opposite holds true.” Still, it is just a conjecture that requires additional studies, and, for this reason, it was not sufficiently stressed in the previous version. We have modified this part of our manuscript. We have also substantially changed the abstract.

**The referee writes:**
>My main criticism about the approach is the following. The authors study numerically chains with L=25 sites up to times of order 10^3. One would expect that to address relaxation one should consider times t\lesssim L, avoiding revival effects. I would appreciate if the authors could address and clarify this point.

In systems with periodic boundary conditions (which we use) the size L can directly determine characteristic time scales in correlation functions only in certain anomalous cases, e.g., in integrable systems with ballistic transport. On the other hand, for perturbed integrable models, the transport is (normal) dissipative and the relevant length scale is the effective mean-free path decreasing with increasing the perturbation strength, g. Provided that the mean-free path is smaller than the system size, correlation functions should become essentially L independent (which we monitor by comparing with results for smaller systems, which we, however, do not display in the manuscript). Therefore the accessible system sizes set the lower bounds mostly on the accessible values of g. Since we are not able to directly evaluate the mean-free path, we use a parameter range for which correlation functions do not reveal any significant L dependence but also exhibit clear quadratic dependence of g. In order to address this problem we have rewritten the paragraph on page 3 that starts with: “As a first step, we establish the range of the perturbation strength,…”
The decay of operators which don’t commute with S^z_{tot} does not show g^2 dependence because the decay originates from a different physical mechanism, i,e. from lifting the degeneracy. In this case, we explicitly discuss that the numerical results do not show any significant L-dependence.

---

## Round 1 · Referee Report · Anonymous (Referee 2) · 2022-4-23

Strengths

1- Very long time numerical data using state of the art methods 2- Interesting problem

Weaknesses

1- The conclusions based on the numerics are plausible, but it seems difficult to know how many conserved quantities really contribute for a given observable a priori. 2- Some points could be clarified.

Report

In this paper, the authors study the late time relaxation of correlation functions in a non-integrable perturbation of the XXZ spin chain. The perturbation is taken to be small, so that the effects of the various local and quasilocal charges of the XXZ chain can still be felt.

The main result compared to previous papers is the appearance of multiple relaxation times, which may be explained in terms of the decay of several conserved quantities back in the integrable system.

Overall this is an interesting paper on an interesting problem. The numerical results are also of good quality. The interpretation of the results is also delicate, but done with a great deal of objectivity. I would tend to recommend publication, provided the points below are addressed. Also, it would be nice if the authors could clarify what is difficult or not when trying to identify the relevant charges needed to explain a given result. For example, an algorithm for determining such relevant charges is mentioned but never used.

Requested changes

Page 2, the last paragraph is not very clear. Another study of the non integrable XXZ chain focusing on anisotropies Delta=0.5 and Delta=1 is mentioned, however, all numerical data in the previous sections is for Delta=0.5.

What is the origin of the oscillations which start appearing at very late time in several plots? Are those numerical artifacts?

Page 11, in the first paragraph, it is mentioned that the observables were carefully selected to have small overlap onto the modes with slow decay. How was this done exactly ? By trying several parameters values and observing the late time results, or from first principles?

I also noticed the following typos:

1) Page 2, second paragraph: 'generalized Gibbs ensable' -> 'generalized Gibbs ensemble' . 'have demonstrate' -> 'have demonstrated'

2) Page 5, after figure 3. 'Above results' -> 'The above results'. Same remark in several other places.

3) Page 6, near the end of the second paragraph. A word is missing after 'the spin-current stiffness'. First line in section 3. 'the of long-time decay' ->'the long-time decay'.

  • validity: high
  • significance: high
  • originality: good
  • clarity: good
  • formatting: excellent
  • grammar: good

Author:  Jacek Herbrych  on 2022-05-21  [id 2502]

(in reply to Report 2 on 2022-04-23)

**The referee writes:**

>It would be nice if the authors could clarify what is difficult or not when trying to identify the relevant changes needed to explain a given result. For example, an algorithm for determining such relevant charges is mentioned but never used.

In the revised manuscript (below Eq. 6) we describe the basic idea of the numerical algorithm and explain why it is applicable only to systems that are much smaller than the systems studied in the present work.

>Page 2, the last paragraph is not very clear. Another study of the non integrable XXZ chain focusing on anisotropies Delta=0.5 and Delta=1 is mentioned, however, all numerical data in the previous sections is for Delta=0.5.

We have newly written this paragraph. We stress that the case Delta=0.5 has been used throughout the manuscript. However, in the last section concerning the decay of operators which do not commute with S^z_{tot} we additionally study the case with Delta=1. In contrast to other commensurate values of Delta, for Delta=1 one may easily find a conserved quantity that does not commute with S^z_{tot}. This simplicity was the motivation for discussing the Delta=1 case in the last section of our manuscript.

>What is the origin of the oscillations which start appearing at very late time in several plots? Are those numerical artifacts?

The time dependence of the correlation functions consists of a smooth part (that is relevant for the thermodynamic limit) and tiny finite-size fluctuations which are unavoidable in an arbitrary finite system. The smooth part decays in time and eventually becomes of the same order of magnitude as the finite-size fluctuations. Since we plot the numerical data on the log-scale, the fluctuations seem to be amplified for a large time.

>Page 11, in the first paragraph, it is mentioned that the observables were carefully selected to have small overlap onto the modes with slow decay. How was this done exactly ? By trying several parameters values and observing the late time results, or from first principles?

In the revised manuscript (page 11) we explain the main idea, which allowed us to tune the overlap. Namely, we made use of the result from Ref. [3] that the projection of spin current on the energy current is proportional to S^z_{tot} thus the overlap may be tuned via changing the magnetization sector.

---

## Round 1 · Referee Report · Anonymous (Referee 3) · 2022-5-8

Strengths

  • challenging numerics on transport in spin chain

Weaknesses

  • the results do not consist in a clear physical effect, or in one clear message, but rather in a sequence of numerical observations. As a consequence, it is unclear whether or not the paper really solves a problem, or whether or not it opens up a clear new research direction.

Report

The paper is scientifically sound, and the numerics is done carefully. I think the paper is interesting for experts in that field, so the results deserve to be published despite the fact that they do not look very surprising.

I doubt the acceptance criteria of Scipost Physics are met though. As far as I can see, the paper neither reports a groundbreaking computational discovery, nor presents a breakthrough on a well-identified problem, nor presents a new research direction with clear potential for future work, nor provides a new link between different research areas.

Consequently, I would recommend to transfer the paper to Scipost Physics Core.

Requested changes

None

  • validity: high
  • significance: ok
  • originality: ok
  • clarity: high
  • formatting: good
  • grammar: excellent

Author:  Jacek Herbrych  on 2022-05-21  [id 2503]

(in reply to Report 3 on 2022-05-08)

Understanding the properties of nearly integrable systems seems to be crucial for establishing the relation between theoretical studies of integrable models and experimentally realizable systems. The essential consequence arising from the integrability-breaking perturbations concerns the relaxation of correlation functions which occurs at a finite time scale. Then, there is an obvious question of whether a single perturbation introduces a single relaxation time or multiple relaxation times. Up to our best knowledge, we provide the first direct numerical evidence for the latter scenario. We also formulate a conjecture which explains why typical observables show exponential decays with a single relaxation time. We believe that this message is important for the ongoing research on integrable and nearly integrable systems.

Motivated by the Referee’s criticism as well as by the first remark of the Referee 1 we have significantly changed the abstract and conclusions. Refer to our response to Report 1 for more details.

---

## Editorial Decision

published